# Stereo-Based Vision and Tactile Sensing for Robust Dual-Arm Robotic Connector Assembly

*Abstract*— The automation of Deformable Linear Object (DLO) manipulation remains a key challenge in industrial production, where tasks such as *connector assembly* are still largely performed manually. Although prior work has demonstrated reliable wire terminal pose estimation and insertion monitoring using vision and tactile sensing, these approaches typically assume a fixed and known connector pose, limiting their applicability in flexible manufacturing scenarios. This paper presents a dual-arm robotic system for fully autonomous *connector assembly*. The proposed setup employs two manipulators: one dedicated to wire perception and insertion, and the other to connector localization and manipulation. A stereo vision system enables robust 6D pose estimation of the wire terminal, while a custom mechatronic gripper with integrated tactile sensing supports accurate insertion monitoring. In parallel, the second arm performs connector grasping. By combining complementary visual and tactile feedback across both manipulators, the system achieves the precision required for tight-tolerance insertion tasks without relying on fixed fixtures. Experimental results demonstrate the effectiveness of the proposed approach in realistic scenarios, highlighting improved robustness and flexibility compared to single-arm solutions.

## I. INTRODUCTION

The rising demand for industrial automation is driving the development of robotic systems capable of handling deformable objects [1]. Among these, Deformable Linear Objects (DLOs), such as cables, wires, and hoses, are critical components in industries like automotive, aerospace, and electronics [2]–[6]. Unlike rigid parts, DLOs exhibit complex behaviors, including bending, twisting, and stretching, that make robust robotic manipulation and perception highly challenging [7], [8]. *Connector assembly* is a particularly demanding task within wire harness manufacturing, requiring the high-precision insertion of crimped metal terminals (*pins*) into designated connector housing slots. Traditionally, this task has relied on human operators who use a combination of visual and tactile feedback to ensure correct alignment and axial orientation [9], as shown in Fig. 1a.

In [10], a methodology to partially automate connector assembly was presented, focusing on the *pin* insertion task. The approach relied on a single robotic manipulator equipped with a custom mechatronic tool integrating a convergent stereo vision system for robust 6D pin pose estimation, along with sensorized fingers for real-time insertion monitoring. However, the connector housing was assumed to be fixed at a known and static position, which limits the level of autonomy and flexibility required in realistic industrial scenarios.

Achieving fully autonomous and robust connector assembly requires overcoming this limitation by enabling both

Supplementary video material is available here.

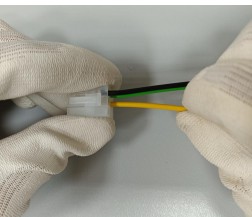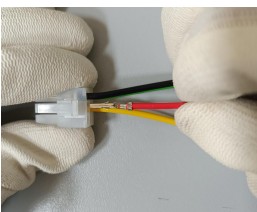

(a) manual insertion by a human operator

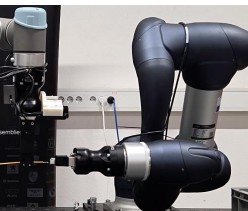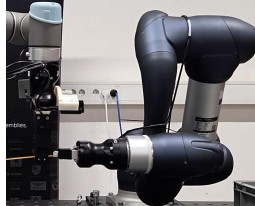

(b) autonomous insertion by the proposed dual-arm system

Fig. 1. Comparison between manual and autonomous connector assembly.

perception and manipulation of the *connector* in addition to the *pin-wire*. To this end, this work adopts a dual-arm configuration, improving over the existing single-arm solutions [10]. One robot is responsible for handling the *pin-wire*, while the second robot is tasked with grasping and positioning the *connector* for the subsequent dual-arm manipulation phase.

This shift introduces new perception and manipulation challenges. In particular, while standard vision-based methods can provide an initial 6D pose estimate of the *connector* sufficient for grasping, their accuracy is typically insufficient for tight-tolerance insertion tasks due to sensor resolution limits and uncertainties introduced during manipulation. To address these challenges, the second arm is equipped with a custom-designed mechatronic tool that enables high-precision in-hand connector pose refinement. This is combined with the stereo-based perception and tactile-guided insertion strategy developed for the *pin-wire*, allowing the system to achieve the accuracy required for reliable insertion. Preliminary experimental results demonstrate that the proposed dual-arm system enables successful connector assembly in realistic scenarios, improving robustness and flexibility compared to the previous single-arm approach.

## II. MECHATRONIC TOOL

The proposed methodology uses a custom mechatronic tool, see Fig. 2, built on a commercial parallel-jaw gripper (Robotiq Hand-E) enhanced with a stereo vision system and sensorized fingers featuring photo-reflector tactile sensors.

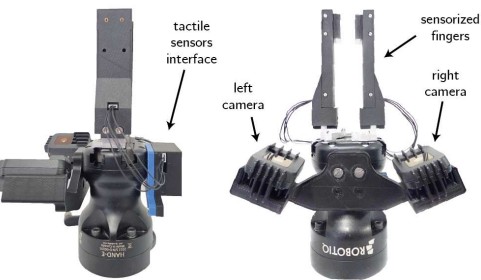

Fig. 2. Mechatronic tool for connector assembly comprising a converging stereo-based camera system and sensorized fingers.

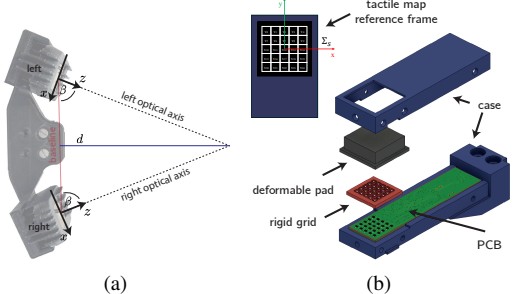

Fig. 3. (a) Geometric setup of the stereo-based vision system. (b) Sensorized fingers with tactile map reference frame.

The geometric configuration of the vision system is shown in Fig. 3a. Unlike conventional stereo setups with parallel camera sensors, a converging arrangement of the sensors is used to enhance accuracy [11]. To balance depth accuracy and minimize the gripper's bulk, the system adopts a camera orientation, indicated with $\beta$ in Fig. 3a, of $68°$ and a baseline of $0.12$ meters. Given the finger dimensions, the distance between the closed gripper fingers and the baseline axis ($d$) is $0.13$ meters. The system employs two Luxonis OAK-1 cameras, offering a resolution of $3840 \times 2160$ pixels at 30 fps. Calibrated for intrinsic, extrinsic, and stereo parameters, they enable accurate depth perception and precise 3D vision essential for the task.

The sensorized fingers employed in this work are based on optoelectronic technology [12]. Figure 3b shows a CAD drawing of the finger setup. The tactile sensors consist of a $5 \times 5$ matrix of sensing points, called *taxel*, each one composed of a photo-reflector, i.e. an LED and a phototransistor optically matched. The sensing matrix is covered by a silicone pad, whose deformations are detected by the photo-reflectors. Indeed, the voltage returned by each taxel depends on the quantity of light emitted by the LED that reaches the phototransistor, which varies according to the extent of the deformation of the silicone layer. The sensor is associated with a $\Sigma_s$ frame, see zoomed tactile map of Fig. 3b.

The wire to be assembled is positioned on a pair of supporting clips, simulating a feeding system that supplies wires one at a time to the robot. It is important to highlight that, due to the wire's flexibility, potential misalignments during grasping, and possible collisions with the environment, the exact poses of both the *pin* and the *connector* holes are not known a priori and must be estimated prior to insertion.

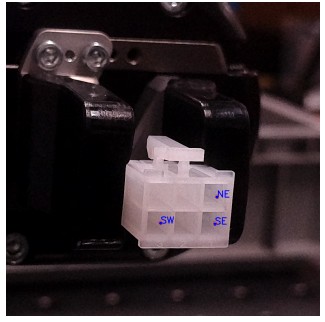 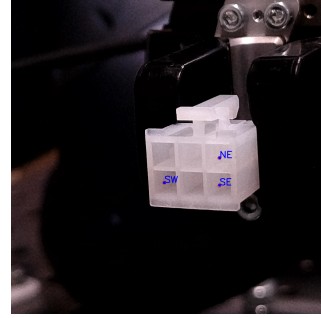

(a) left view          (b) right view

Fig. 4. Detection of connector holes from stereo views using the YOLO-based model, achieving robust and precise in-hand connector pose.

## III. METHOD

### A. Connector Pose Estimation

The first phase focuses on the manipulation of the *connector* by the left robotic arm. Given the complexity of the assembly, high-precision localization is required. Firstly, the camera mounted on the wrist of the right arm provides the initial scene perception. The connector's 6D pose is estimated using DOPE [13]. This model is trained on synthetic datasets rendered from the CAD model of the connector, allowing it to generalize across various lighting conditions and viewing angles.

While DOPE provides a reliable coarse pose estimate suitable for planning the initial grasp, its accuracy is insufficient to ensure successful insertion. Consequently, an additional refinement stage based on hole detection is introduced to achieve the sub-millimetric precision and robustness required by the task. The process is illustrated in Fig. 4.

In this refinement stage, the left arm positions the connector such that its insertion-side surface is clearly visible to the right arm's camera system. A YOLO-based detection method is then applied to identify the six connector holes, using a model trained on synthetic data. Detection is performed on both stereo images. Corresponding detections across the two views are matched and triangulated to reconstruct the 3D coordinates of the hole centers. Finally, the full connector pose is estimated by exploiting the known rigid geometric relationship among these points.

### B. Stereo-Based Pin Pose Estimation

Simultaneously, the right robotic arm estimates the 6D pose of the wire *pin* from a pair of stereo images capturing the terminal region. The vision system processes these images through a multi-head neural network (see Fig. 5), which predicts both the image coordinates of the *tip* and *tail* keypoints in the two views, as well as the axial rotation of the pin around its longitudinal axis.

The predicted 2D keypoints are triangulated using the known stereo geometry to reconstruct their corresponding 3D positions in the world frame. These reconstructed *tip* and *tail* points define the main axis of the pin, while the estimated axial angle provides the rotation around this axis. By combining these components, the full 6D pose of the

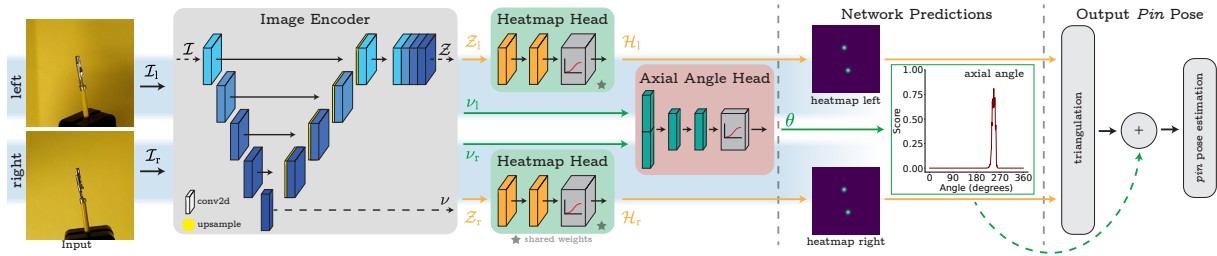

Fig. 5. Overview of the stereo-based neural network architecture and the triangulation process for estimating the *pin* pose.

pin is obtained, enabling accurate alignment with the target connector hole.

Rather than directly regressing the full pose, the proposed approach decomposes the problem into intermediate predictions that are easier to learn and more robust in practice. In particular, the network estimates 2D keypoints and the axial angle from stereo images, which are then fused geometrically through triangulation. This strategy avoids the need for large-scale datasets with full 6D pose annotations (difficult to obtain in practice) and allows efficient use of real-world data.

The network architecture consists of two main components. First, an image encoder based on a lightweight ResNet-like backbone extracts rich multi-scale features by aggregating information from multiple residual blocks. This results in both spatial feature maps, used for precise keypoint localization, and compact feature vectors, used for global reasoning tasks such as angle estimation.

Second, a keypoint prediction head operates on the spatial features to generate heatmaps corresponding to the *tip* and *tail* in both stereo views. The weights are shared between the left and right branches to enforce consistency and reduce overfitting. Keypoint locations are obtained by selecting the maxima of the predicted heatmaps, while training relies on Gaussian-shaped supervision targets and a mean squared error loss.

Finally, the axial angle estimator leverages the feature vectors extracted from both views, combining them to exploit complementary stereo information. A sequence of fully connected layers predicts a discretized distribution over possible angles, allowing the model to capture uncertainty and ambiguity in the estimation. This prediction is trained using a binary cross-entropy loss.

### C. Tactile-Based Pin Insertion

A tactile sensing strategy is adopted to directly capture contact interactions at the grasped object. A contact indicator $\mu_c = (\mu_x, \mu_y)$ is computed quantifiyng the displacement of the contact distribution over the sensor surface. In particular, the centroid of the tactile response is estimated from the taxel readings, and its variation with respect to a reference configuration provides a measure of contact evolution. For the insertion task, only the $\mu_x$ component is considered, as it aligns with the primary direction of motion.

The tactile signal $\mu_x$ is analyzed in real time to detect key interaction events during the task execution (see Alg. 1). Specifically, three conditions are identified: *absence of significant contact*, completion of the *push phase*, and

---

**Algorithm 1:** Tactile-Based Pin Insertion Algorithm

**Input:** tactile indicator $\mu_x$, max displacements $\delta_{\text{push}}$, $\delta_{\text{pull}}$, classifier model
**Output:** boolean success flag `success`
`success` ← False
$x_{\text{push}} \leftarrow 0$, $x_{\text{pull}} \leftarrow 0$
**while** $x_{push} < \delta_{push}$ **do**
    $x_{\text{push}} \rightarrow$ `UpdateRobotPosition()`
    $\mu_x \leftarrow$ `ComputeTactileIndicator()`
    `class` ← `classifier.predict(`$\mu_x$`)`
    **if** $class = 1$ **then**
        `StopRobotMotion()`
        **break**

**while** $x_{pull} < \delta_{pull}$ **do**
    $x_{\text{pull}} \rightarrow$ `UpdateRobotPosition()`
    $\mu_x \leftarrow$ `ComputeTactileIndicator()`
    `class` ← `classifier.predict(`$\mu_x$`)`
    **if** $class = 2$ **then**
        `StopRobotMotion()`
        `success` ← True
        **break**

**return** `success`

---

completion of the *pull phase*. These events exhibit distinctive patterns in the tactile signal, which remains close to zero during free motion, increases in one direction at the end of insertion, and in the opposite direction when the pin is correctly locked and pulled back.

Based on this observation, the insertion strategy is formulated as a two-stage procedure. During the push phase, the robot advances until a *push completed* event is detected, indicating either successful insertion or contact due to misalignment. Subsequently, a pull motion is executed to verify locking: if a *pull completed* event is detected, the insertion is considered successful; otherwise, the task is classified as failed.

To enable reliable classification of these events, a machine learning model is trained on tactile data collected during a few task executions. The robot performs controlled push and pull motions while recording the $\mu_x$ signal. Both successful and failure cases are included by either using the vision-based alignment or deliberately introducing misalignment. The resulting signals are labeled according to the detected events. Data augmentation techniques, such as noise injection

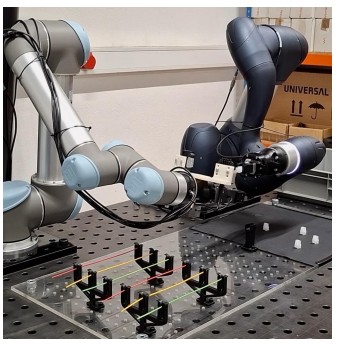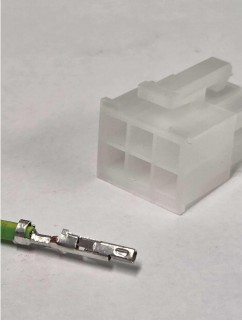

Fig. 6. Experimental setup: overview of the dual-arm robotic platform (left); detailed view of the connector and the pin (right).

and amplitude scaling, are applied to improve robustness. A $k$-Nearest Neighbors (KNN) classifier is then trained to distinguish among the three event classes. During execution, the classifier operates on sliding temporal windows of the tactile signal, enabling continuous and real-time detection of contact transitions. This approach provides a simple yet effective mechanism to monitor the insertion process and ensure reliable task completion.

## IV. EXPERIMENTS

### A. Full Task Evaluation

The proposed methodology is validated through a series of preliminary assembly experiments aimed at assessing the accuracy and robustness of the dual-arm robotic strategy. The system is implemented within the ROS2 framework, which manages robot control, stereo vision processing, and tactile data acquisition.

The experiments were conducted on a workstation equipped with an Intel i7-9700 CPU and an NVIDIA GTX 1660Ti GPU. The system operates at a processing rate of 5 Hz for vision-based pose estimation and 500 Hz for tactile feedback.

The evaluated connector features six holes of size $3.7 \times 3.7$ mm, while the pin has dimensions of $1.6 \times 1.6 \times 14.8$ mm. The experimental setup consists of a dual-arm robotic platform composed of a UR5 and a Doosan manipulator (see Fig. 6).

The UR5 features the mechatronic tool of Fig. 2, while the Doosan robot assists in connector grasping (i.e. left arm).

Preliminary results indicate that the dual-arm configuration, combined with the proposed in-hand connector pose refinement, allows both robustness and precision. The system consistently achieves the accuracy required for successful insertion, demonstrating its effectiveness for fine assembly tasks.

### B. Analysis of Axial Angle Confidence

The axial angle prediction head of Sec. III-B provides not only an estimate of the angular orientation, but also a confidence value associated with that estimate. Indeed, the network outputs a discretized Gaussian-like distribution over the axial angle domain. The predicted angle corresponds to the peak of this distribution, while the peak amplitude

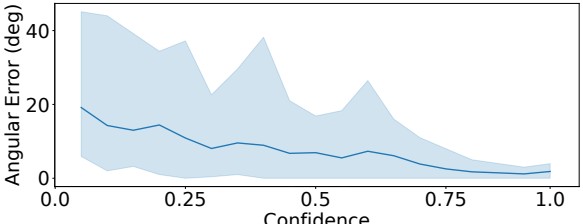

Fig. 7. Relationship between predicted confidence and axial angle estimation error. Higher confidence predictions exhibit lower mean error and smaller variance, indicating an inverse correlation.

is interpreted as a confidence score. In this way, a sharp and pronounced distribution indicates a reliable prediction, whereas a flatter profile reflects greater ambiguity in the estimated orientation.

To assess the reliability of this confidence measure, its relationship with the angular prediction error is analyzed across test samples collected during insertion experiments. The samples are grouped according to their predicted confidence, and, for each confidence interval, the mean angular error and standard deviation are computed. The corresponding results are reported in Fig. 7.

The analysis reveals a clear inverse correlation between confidence and angular error. In particular, predictions associated with high confidence exhibit low mean angular errors and limited variability, indicating that the network is generally reliable when the output distribution is sharply peaked. Conversely, low-confidence predictions are associated with larger errors and higher dispersion, which is consistent with visually ambiguous cases such as reflections, reduced contrast, or less distinctive pin appearances. These results indicate that the confidence score produced by the axial angle prediction head is informative of the actual quality of the estimate. For the pin insertion case analyzed in this work, this is especially relevant given the tighter geometric tolerances of the connector, where even moderate angular inaccuracies may compromise successful insertion. Therefore, the predicted confidence can serve as a useful indicator for identifying uncertain pose estimates and, potentially, for triggering additional verification or recovery strategies.

## V. CONCLUSIONS

This work presented a dual-arm robotic system for fully autonomous connector assembly involving deformable linear objects. By extending previous single-arm approaches, the proposed method enables simultaneous perception and manipulation of both the pin-wire and the connector, removing the need for fixed fixtures and increasing system flexibility.

Preliminary experimental results demonstrate that the proposed dual-arm configuration achieves the accuracy and robustness required for tight-tolerance assembly tasks. The system successfully integrates visual and tactile information across both manipulators, improving performance compared to previous single-arm solutions.

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
