# OpenReview forum: "Stereo-Based Vision and Tactile Sensing for Robust Dual-Arm Robotic Connector Assembly"
_IEEE.org/ICRA/2026/Workshop/Manipulation_Robustness — ICRA 2026_

### Official Review · Reviewer_HcWm · 2026-05-16
**Dual-Arm Connector Assembly System**

**Rating:** 7
**Confidence:** 3

**Review:**

The paper presents a well-engineered dual-arm robotic system for autonomous connector assembly using stereo vision and tactile sensing. The overall system integration is strong, and the industrial motivation is realistic. I especially like the stereo-based pin pose estimation pipeline and the practical use of tactile sensing for insertion monitoring.

pros
- Strong system integration for a realistic industrial assembly task.
- Clear practical relevance for DLO and wire harness manipulation.
- Effective stereo-based pose estimation and tactile-guided insertion.
- Dual-arm setup improves flexibility over fixed-fixture approaches.

cons
- Task setting is still relatively constrained.
- Limited robustness evaluation and ablation studies.

---

### Decision · Program_Chairs · 2026-05-21

Accept